# The Preparation and Clinical Efficacy of Amnion-Derived Membranes: A Review

**DOI:** 10.3390/jfb14100531

**Published:** 2023-10-20

**Authors:** Alison L. Ingraldi, Robert G. Audet, Aaron J. Tabor

**Affiliations:** 1Carmell Corporation, Pittsburg, PA 15203, USA; raudet@axobio.com; 2Department of Research and Development, Axolotl Biologix, Flagstaff, AZ 86001, USA; 3Department of Biological Sciences, Northern Arizona University, Flagstaff, AZ 86011, USA; 4Department of Clinical Operations, Axolotl Biologix, Flagstaff, AZ 86001, USA

**Keywords:** human amniotic membrane, amnion, chorion, amnion–chorion, processing, regenerative medicine, skin substitute, chronic wound, anti-inflammatory, clinical application

## Abstract

Biological tissues from various anatomical sources have been utilized for tissue transplantation and have developed into an important source of extracellular scaffolding material for regenerative medicine applications. Tissue scaffolds ideally integrate with host tissue and provide a homeostatic environment for cellular infiltration, growth, differentiation, and tissue resolution. The human amniotic membrane is considered an important source of scaffolding material due to its 3D structural architecture and function and as a source of growth factors and cytokines. This tissue source has been widely studied and used in various areas of tissue repair including intraoral reconstruction, corneal repair, tendon repair, microvascular reconstruction, nerve procedures, burns, and chronic wound treatment. The production of amniotic membrane allografts has not been standardized, resulting in a wide array of amniotic membrane products, including single, dual, and tri-layered products, such as amnion, chorion, amnion–chorion, amnion–amnion, and amnion–chorion–amnion allografts. Since these allografts are not processed using the same methods, they do not necessarily produce the same clinical responses. The aim of this review is to highlight the properties of different human allograft membranes, present the different processing and preservation methods, and discuss their use in tissue engineering and regenerative applications.

## 1. Introduction

The unique, multilayered structure of the human amniotic membrane (AM), and its associated biological and physical characteristics, make it a highly biocompatible material suitable for use in a variety of regenerative medicine applications [1]. Human AM has been used for over a century in a wide variety of clinical applications including ophthalmologic, chronic wounds, burns, plastic reconstruction and periodontal procedures [2,3]. The positive outcomes from the historic uses of AM prompted further research and clinical investigations into additional human therapeutic applications. As research supported the use of AM for anti-inflammatory, angiogenic, anti-angiogenic and antimicrobial properties, and a source of diverse growth factors, more clinically relevant applications have been demonstrated [4,5,6].

Fetal membranes were first used in the 1900s to successfully treat acute and chronic traumatic wounds, burns, ulcers, and as a novel skin substitute for grafting [7,8,9,10]. Early uses in the ophthalmic field addressed various ocular pathologies, including plastic surgery of the conjunctiva [11] and ocular burns [12]. The amniotic membrane’s inherent elasticity allows it to conform to complex contoured surfaces, which led to its application in many reconstructive procedures, including the creation of artificial vaginas, and treating chronic complex wounds in diabetic patients [2,13,14]. Applications have expanded to support every tissue type within the human body, including ophthalmology; periodontal; skin and wound applications; burns; craniotomies presenting with dural defects; bladder; genital, and peritoneal reconstructive surgeries; prevention of surgical adhesions in abdominal surgeries; tendon and ligament repair; articular joint repair; nerve wrapping; chronic wounds; burns and plastics [2,3,14,15,16,17,18,19]. The growing fields of tissue engineering and regenerative medicine have brought greater attention to the varied human amniotic membrane as a tunable matrix for tissue engineered constructs. This review aims to provide a background on the anatomical and physiological properties of the amniotic membrane, the processing and preservation methods that produce the clinically available products and highlight the applications in four of the most common regenerative medicine fields of application.

## 2. Anatomy and Physiology of the Amniotic Membrane

The human placenta is a highly specialized hemochorial villous organ, comprising fetal and maternal membranes and amniotic fluid, which support the normal growth and development of a fetus [20]. The fetal membranes surround and protect the fetus throughout pregnancy and eventually undergo organized rupture during the first stage of labor. Placental tissue comprises three key tissue layers: the maternally derived decidua and the embryo derived amnion and chorion layers (Figure 1) [20,21,22,23]. The amnio chorionic membrane separates the fetus from the endometrium and forms a fluid-filled sac that protects the fetus from mechanical stresses, providing nutrients, oxygen, and waste removal during intrauterine development [23]. The amnion and chorion membranes can be as large as 1500 cm^2^, however, there is natural variation in the shape and size of placental cells among donors [24]. Next, we discuss the amnion, the membrane layer furthest from the maternal tissue and immediately adjacent to the fetus.

### 2.1. Amnion

The human amniotic membrane (hAM) is sourced from the inner and outer layers of the amniotic sac and comprises two distinct, connected membranes: amnion and chorion. The smooth inner amnion membrane is a thin, tensile, avascular, semi-transparent structure without nerve innervation, muscle, or lymphatic vessels, with a reported relative thickness of 20–500 µm [24] or 35–60 µm [23]. It comprises three major histological structures: an epithelial monolayer, a thick basement membrane, and an avascular stroma in contact with the underlying chorion. The innermost epithelium layer, facing the fetus, in direct contact with the amniotic fluid, consists of a single layer of cuboidal epithelial cells with apical microvilli uniformly arranged on the basement membrane, which help in the absorption and secretion of solutes and water [14,20,23,25]. These multipotent amniotic epithelial cells (AECs) express stem cell-specific transcription factors, including octamer-binding protein-4 (Oct-4), SSEA-4, and NANOG [26,27]. The basement membrane is one of the thickest membranes found in all human tissue, provides the necessary support throughout a fetus’s gestation and consists of lamina lucida and lamina densa. These layers contain collagen types III, IV, V and VII, laminin-1, laminin-5, fibronectin, and various growth factors associated with cell differentiation and survival [22,28]. The avascular stromal layer can be further subdivided into three layers: the compact layer, fibroblast layer and the spongy intermediate layer that separates the amnion membrane from the chorion membrane. The compact layer is a thin acellular layer of strong, reticular fibers containing collagen types I, III, V and VI, and fibronectin. The fibroblast layer consists of collagen types I, III, and VI, fibronectin, laminin, and nidogen. This layer includes fibroblast-like multipotent mesenchymal stromal cells (AMSCs), which secrete growth factors, cytokines, and matrix proteins, which help to provide the mechanical integrity of the amnion and monocyte-like Hofbauer cells [22,25,29]. The deeper spongy layer, also called the zona spongiosa, comprises a network of type III collagen, mucin, reticulin, hydrated glycoproteins, and proteoglycans and may also contain fibroblasts and Hofbauer cells, which may assist in embryonic cell differentiation. This spongy layer is loosely connected to the chorion, allowing the amnion and chorion to slide against each other, and can easily be separated by blunt dissection [15,21,25]. The second membrane, tightly opposed to the amnion, is the chorion.

### 2.2. Chorion

The external chorion membrane is a thin, opaque, fibrous membrane connected to the amnion and the outer decidua. The chorion layer is three to four times thicker than the amnion layer; the thickness varies from 20 to 200 µm, with a weaker tensile strength than amnion, and consists of a reticular layer, pseudo-basement membrane and trophoblast cell region, firmly adhered to the maternal decidual tissue at term [22,23]. The reticular layer contains a transition of collagen types I, III, IV, V, VI with proteoglycans on a pseudo-basement layer of collagen type IV, fibronectin, and laminin [22,28]. The cytotrophoblast shell of the deepest chorion layer comprises cubical or prismatic cells, which form chorionic villi facing towards the inner (fetal) surface, and functions to provide nutrition to the developing embryo protected within [22,23,30]. The chorionic villi increase the surface area of the placenta and comprise multinucleated cytotrophoblast cells, which contain the same genetic material as the fetus. As a result, cells of the chorionic villi can be collected and examined to determine any fetal genetic disorders in a procedure known as chorionic villus sampling. The placenta has a maternal component, the basal plate derived from the decidua basalis, and a fetal portion called the chorionic plate, formed from the chorion frondosum. The villi of the chorion frondosum attach to the decidual tissue and secure the fetal placenta to the basal plate with a blood-filled space between where trophoblastic lacunae become the intervillous spaces. Upon completion of the third trimester during gestational development, chorionic villi senesce, leaving the chorionic trophoblasts [23]. Each of the two amniotic membrane layers contain cells which provide the secretory functions of the AM.

### 2.3. Amniotic Membrane Cells

The amnion contains two cell types, from different embryological origins, with characteristic properties of stem cells. Human amnion epithelial cells (AECs), which are firmly connected to the thicker basement membrane and face the amniotic fluid compartment, and human amnion mesenchymal stromal cells (AMSCs), are derived from the fibroblast layer of the membrane. These two multipotent cell types are able to differentiate into various tissue types with potential in organ regeneration. The AMSCs differentiate into classical mesodermal lineages, osteoblasts, chondrocytes, and adipocytes, as well as to the three primary germ layers representing cardiomyocytic, myocytic, endothelial, neural, and hepatocytic lines [5,29]. The AECs are capable of differentiating into adipocytes, osteocytes, nerve cells, cardiomyocytes, myocytes, hepatocytes, hematopoietic cells, endothelial cells, kidney cells, and retinal cells [31]. The chorion region cell population consists of human chorionic mesenchymal stromal cells (CMSCs), and human chorionic trophoblastic cells (CTCs) [29]. The AM has a dual secretory role, providing nutrients and growth factors to the amniotic cavity and providing critical feedback to the maternal decidua to support the developing fetus [20]. These soluble factors include hyaluronic acid (HA), tissue inhibitors of metalloproteinases (TIMPs), interleukins (ILs), migration inhibitory factors, and prostaglandins, all of which contribute to the unique applications and capabilities of using human amnion and chorion byproducts in wound care and tissue engineering [5,30,32,33,34]. See Table 1 and the following section for a summary of the key properties and functions of the amniotic membrane.

### 2.4. Amniotic Membrane Properties

Research efforts over the last few decades have led to the discovery of the different characteristics and capacities inherent in amnion and chorion membranes, such as anti-inflammatory, immunomodulatory, anti-scarring, antimicrobial, angiogenic, and cell recruitment leading to tissue repair and regeneration. The two major components of the placental membranes which support these functions are the cells and the extracellular matrix (ECM), and the biological influence they have on each other. The cells are responsible for the synthesis and turnover of the ECM components, secretion of growth factors and cytokines and regulation of the fetal and maternal environment, while the ECM significantly influences the cells’ behaviors, such as growth, adhesion, and differentiation in vivo. A study by Koob and colleagues provided details on the individual cytokines and growth factors provided by each layer and found the chorion layer to contribute higher percentages of platelet-derived growth factor (PDGF-AA) and vascular endothelial growth factor (VEGF) than the amnion layer [35].

The immunomodulatory properties of the AM, conferred by the presence of HLA-G and absence of MHC Class I surface markers, contribute to its immunosuppressive function and reduce the risk of rejection as an allogeneic tissue. Anti-inflammatory properties in the AM and AM cells contribute to a shift in macrophages from M1 proinflammatory towards M2 anti-inflammatory subtypes, and a shift to anti-inflammatory growth factors and cytokines, which contribute to a T-cell shift from the Th1 to Th2 phenotype with an increase in suppressor T-regulatory populations [36,37].

There are conflicting data regarding the angiogenic potential of the AM. Hao et al., identified the mRNA of anti-inflammatory (IL-1ra and IL-10) and anti-angiogenic proteins (endostatin, TIMP-1, -2, -3, -4, TSP1) in isolated AECs and AMSCs as well as the presence of these proteins in hAM [38]. In a rat dorsal skinfold chamber model, the amnion epithelial side when opposed to the wound inhibited angiogenesis compared to the mesenchymal side [39]. Amniotic MSCs demonstrate angiogenic activity assessed by RT-PCR, endothelial cell tube forming assays and recovery of hindlimb ischemia [40].

The biomechanical properties of the amnion make it resistant to various proteolytic factors due to the interstitial collagen layers [14]. The membrane also reduces bacterial infiltration and contamination of wounds by acting as a barrier to reduce the accumulation of microbes within a wound [28]. The presence of elastin makes amnion more elastic, tensile, and versatile than chorion membrane, while the chorionic bilayer of protein fibers makes it a tougher membrane, providing more support. Amniotic membrane facilitates the proliferation and migration of epithelial cells with an abundance of growth factors and cytokines, a beneficial property utilized in wound healing applications. The basement membrane and ECM structures provide a suitable support for epithelial cell growth, retaining permeability to oxygen and nutrients, in contrast to synthetic materials [3,4,28,41]. Not only does this material encourage cellular growth and differentiation in various damaged tissue spaces, but it is also possible to use these membranes as a biological substrate to seed and culture specific cell types [3,42]. The immunomodulatory properties of amniotic cells, in part due to their lack of expression of MHC Class I and II markers and expression of HLA-G, make this a diverse material that increases the chances of successful tissue grafting [43].

Antimicrobial peptides are expressed within the amnion and chorion during pregnancy to protect the fetus from and respond to maternal infection. The amniotic AECs and AMSCs express an array of these peptides, including α and β-defensins, elafin, LL37, secretory leukocyte protease inhibitor (SLPI) and other antimicrobial peptides [30,41,44,45]. A study by Kjaergaard et al., tested the in vitro antimicrobial effects of fresh amnion and chorion against specific microorganisms on agar plates and in liquid culture and demonstrated that the membranes had an inhibitory effect on the microorganisms in agar plates, but a liquid culture of chorion demonstrated a marginal effect [44]. In a study by Ramuta and colleagues, a disk diffusion study of several gram-positive and gram-negative bacteria cultured with fresh or frozen and thawed hAM demonstrated no diffusible antimicrobial activity, however the homogenates demonstrated antibacterial activity against several bacterial strains confirming that antimicrobial activity is present within the tissue [46].

The presence of antimicrobial peptides (AMPs) can positively impact surgical outcomes by reducing or preventing microbe growth and establishing biofilms during the healing process. AMPs have been identified in fresh, lyophilized, and cryopreserved hAM [41,45] and dehydrated human amnion–chorion membrane (dhACM) [47]. However, most antimicrobial activity in disk diffusion assays was observed in fresh or viable cryopreserved membranes, confirming that viable epithelial cells actively secrete these peptides [45].

The therapeutic applications that benefit from the biological utilities and properties of amnion and chorion have been identified and discussed within the tissue engineering community; however, further research and development is needed to verify these properties, understand how the processing methods affect these benefits and what further enhancements could be employed.

In general, the amnion and chorion have demonstrated beneficial biological and mechanical properties for regenerative medicine and tissue engineering applications; however, these properties may vary depending on the processing method employed.

## 3. Amniotic Membrane Processing

### 3.1. Preparation

Human amnion-derived membranes are collected from consenting mothers following elective cesarean-section delivery under sterile conditions. Birth tissue from normal vaginal delivery has historically not been recommended because vaginal flora contributes to an increased bioburden and a decrease in the tensile strength of the membranes associated with epithelial to mesenchymal transition (EMT) during labor [3,30,48,49,50,51]. Since most birth tissue is irradiated, it is possible to use vaginal placenta, however, bioburden tests and speciation are critical to prevent the possibility of communicable disease transmission [52]. The placenta is then transferred to a tissue bank that follows the Code of Federal Regulations, 21 CFR Part 1271, using a suitable sterile transport medium, such as physiological saline, with or without antibiotics following the tissue bank’s requirements [3,52]. The tissue bank’s transport container with the tissue must be clearly labelled with the donor’s identification details and transported at +2–8 °C until processing occurs [52,53]. Donors are screened to determine social, behavioral, environmental and travel risks. A blood sample is collected for serology, including human immunodeficiency virus types I and II, human T-cell lymphotropic virus, human hepatitis virus types B and C, and syphilis. Additional testing may be performed for cytomegalovirus (CMV), toxoplasma, tuberculosis, Creutzfeldt-Jakob disease, or other infections per regulatory guidelines. Additionally, swabs of the tissue may be collected for a 7-day bioburden test. Donors identified as high risk or with positive serologies are excluded from further processing to avoid the risk of transmissible infections to the recipient [3,50,54,55].

Tissue is typically processed within a cleanroom, biosafety cabinet, or laminar flow hood. If the whole placenta is provided, the amnion–chorion is dissected from the placenta and washed with phosphate buffered saline or similar wash buffer containing antibiotics to remove any blood clots and debris [22,49]. Following the initial wash, the amniotic membrane is separated from the chorion via blunt dissection to facilitate additional processing of each membrane as either amnion, chorion, or amnion–chorion allografts.

Placental tissue’s unique biological and mechanical composition has led to the manufacture of different types of allograft membrane products. Separating the amnion–chorion layer from placental tissue enables the production of either hAM allografts or human chorion membrane (hCM) allografts. The process of laminating the amnion and chorion tissue layers together produces a human amnion–chorion membrane (hACM) allograft. Patented procedures of gently cleaning and laminating the amnion and chorion, followed by dehydration, produce dehydrated hACM (dhACM), e.g., EpiFix^®^ [56] or NuShield^®^ [57]. A different type of dehydrated, bi-layered amnion–amnion graft (dhAAM) is prepared by laminating or fusing the stromal sides of the amnion together then dehydrating, e.g., Axolotl DualGraft™ and dehydrated binate amniotic membrane (DBAM) [58]. A tri-layered amnion–chorion–amnion allograft is manufactured by Integra Life Sciences (Princeton, NJ, USA), AmnioExcel^®^ Plus [59].

Amnion membrane products, including fresh, cryopreserved, lyophilized, dry or wet, intact or decellularized, sterilized or crosslinked, are processed to meet the needs of the clinician, researcher, manufacturer, regulatory designation, or target application [6,34]. Various preparation, preservation, crosslinking, and sterilization methods are selected to match the specific applications and storage requirements. Collected AM may be stored for short periods of time cryopreserved at −80 °C or fresh at +4 °C [60].

To date, various methods have been utilized to prepare, preserve, decellularize, crosslink, and sterilize AM (see Figure 2). Each of these methods can affect the morphological, physio-chemical, mechanical, and biological properties of the amniotic membrane. The differences in membrane processing methods result in the diversity of the membrane products produced and of the outcomes in research studies, clinical trials, and clinical applications.

Amnion-derived membranes may be prepared from fresh or frozen placental tissue; however, in some countries, due to regulatory requirements and the window period of HIV infection, AM must be preserved until a repeat negative HIV test result is obtained after six months [33,54,55,61].

When using fresh AM grafts, immediate transplantation is optimal but not always possible. Lyophilization, dehydration, and cryopreservation processes have been developed to facilitate AM stability and extend shelf life when immediate allograft transplantation is not available [3,62]. The different handling procedures and methods of preservation affect the morphology, transparency, thickness, biochemical composition, protein, and growth factor contents of AM (Figure 3).

For some clinical applications, another processing technique is used. Decellularizing AM provides an ECM for specialty applications, such as ocular surface reconstruction, and for creating scaffolds for tissue engineering and regenerative medicine applications [30,33,63,64,65]. Decellularization describes the process of chemically, biologically, or physically removing native cells from living tissues, leaving the three-dimensional, acellular ECM scaffold [66,67,68]. A decellularized scaffold exposes the native 3D structure, mechanical properties, and the unique protein and growth factor composition of the ECM to promote tissue regeneration and healing. By removing the major immunogenic cellular components, membrane-associated antigens, and soluble proteins, a cell-mediated or humoral immune rejection response after clinical applications is prevented [66,68,69]. Decellularized hCM preserves the mechanical properties and the main ECM protein components, which support different tissue regeneration strategies [69].

There are different chemical, physical, and enzymatic strategies and combinations used to decellularize tissues; however, there is no “gold standard” method, because the removal of cells depends on tissue type [68,70]. Agents such as sodium dodecyl sulphate (SDS), Triton X-100, ethylenediaminetetraacetic acid (EDTA), urea, or trypsin have been used to decellularize amniotic tissue [33,54,64,66,67]. After being exposed to chemical or enzymatic agents, mechanical scraping may be used to remove residual cells, an operator-dependent technique that may be difficult to reproduce and can cause damage to membrane integrity [33,34,54,69]. The efficacy of decellularization also depends on the source, composition, and density of the tissue, as well as the specific layer being decellularized, e.g., amnion or chorion. Decellularization may decrease the thickness, mechanical properties, and the immunogenicity of hAM or hCM allografts to varying degrees depending on the process, technique, and the skill of the technician [70,71].

The relevance of using intact or decellularized AM depends on the application. Improved cell growth has been demonstrated using de-epithelialized AM for culture and cell delivery systems in ocular applications [72,73,74]. Epithelialized membrane is used as a graft in ocular repair applications with the epithelial side intact to facilitate re-epithelialization [50]. The relevance of applying AM with an intact versus a decellularized epithelium is best addressed by the treating physician. The decellularization process is paired with the different preservation techniques described in the next section.

### 3.2. Preservation

Cryopreservation is the most common preservation method for producing viable amnion and chorion membrane products and has been reported by numerous experimental and clinical studies to be safe and efficient [3,75,76,77], though in some instances, cells were found to be nonviable post thaw [63,78]. The standard method of cryopreservation includes the addition of additives such as DMSO or glycerol rather than freezing untreated membrane [33,63,79]. Wagner et al. observed that cryopreservation in a glycerol solution 1:1 with Dulbecco’s modified Eagle medium (DMEM/F-12) did not affect the tensile strength or histology of the hAM when compared to fresh [79]. However, Fénelon et al. observed that hAM cryopreserved in a similar glycerol solution, 1:1 with RPMI had a decreased Fmax (maximum force at failure) compared to fresh [65]. To date we have not found studies addressing the impact of DMSO cryopreservation on mechanical properties.

Cryopreservation does have limitations, which may include the loss of certain growth factors and cytokines. Additional limitations include rinsing cryoprotectants from the tissue prior to application and the potential cytotoxicity of residual cryoprotectants like glycerol and DMSO [65]. It is important to note that the thickness of hAM and hCM varies significantly between maternal donors as well as the preservation procedure used. Storage time of grafts depends on the storage temperature and country-specific regulations [34].

Since cryopreserved AM has a requirement of space-consuming freezer units (from −80 °C to −195 °C) that may not be readily available in clinical sites, other techniques for long term preservation have been developed, including lyophilization and dehydration. In lyophilization, water is removed from the tissue by sublimation. Lyophilized AM can then be stored long-term at room temperature and transported easier than cryopreserved products [80,81]. The protocol generally involves spreading the membrane tissue onto nitrocellulose membrane, cutting to desired sizes, freezing at −80 °C then lyophilizing [34,80]. This process induces some alterations in hAM’s structure and biological and physical properties, and a decrease in autolytic enzyme damage to cells [3,80].

When comparing cryopreservation to lyophilization, cryopreservation resulted in better preservation of proteins and growth factors, whereas lyophilization maintained antimicrobial activity and the morphological structure of the AM [41,80]. Pretreatment with trehalose, a nonreducing disaccharide, found in high concentrations in a wide variety of organisms capable of recovering from near complete dehydration, confers desiccation resistance to cells in the AM by replacing some of the cellular water, stabilizing and protecting the cellular membrane and proteins during the freezing process [33,82]. Fénelon et al., compared the mechanical properties of fresh, cryopreserved, lyophilized/gamma sterilized, and decellularized/lyophilized/gamma sterilized human amniotic membrane and found the different preservation methods caused significant changes in the thickness of hAM in which cryopreservation led to an uptake of hydrophilic glycerol and water with significant swelling of the membrane, while freeze-drying resulted in a loss of liquid, thinning and subsequent restoration of normal thickness upon rehydration [65]. Another comparative study, performed by Tehrani et al., revealed that the preservation methods of cryopreservation and freeze-drying do not affect the antibacterial activity of the AM in comparison with fresh AM and the antibacterial activity is dependent on the bacterial strains [41]. A recent study by Jacob et al., assessed the viability of cryopreserved hCM versus lyophilized hCM preincubated in a trehalose-containing lyoprotectant solution and found comparable cellular viability between tissues preserved by each method. However, viability in both methods was lower than in fresh hCM. They demonstrated comparable native hCM structure following thaw or rehydration, anti-inflammatory and angiogenic activity. Viable lyopreserved hCM prevented adhesion formation in a rabbit abdominal adhesion model [76].

An alternative preservation method used to remove water content from tissue is a temperature-controlled dehydration method. Drying conditions vary and can either be maintained under a heat source or by refrigeration at 4–8 °C [83]. Processed, dehydrated hACM has been shown to retain the biological activity of growth factors and regulatory proteins that play roles in cell proliferation, inflammation, tissue remodeling, and recruitment of stem cells [57,84,85,86]. Controlled dehydration has been used for the various AM layers: amnion, chorion, or amnion–chorion.

The inclusion or exclusion of layers of the amnion tissue within the created graft may affect the growth factor and cytokine content as well as the graft structure [57]. Lei et al. demonstrated the presence and activity of growth factors, proteases and inflammatory cytokines and chemokines in micronized dehydrated hACM [87]. In a study to compare the therapeutic potential of dehydrated and cryopreserved hAM and hACM grafts, Cooke et al. revealed that the extracellular matrix structure in dehydrated hAM and hACM was compromised as compared to cryopreserved hAM and hACM, and that the dehydrated membranes contained lower levels of hyaluronic acid [88]. Controlled temperature dehydration of hAM at 4–8 °C improved the transparency or optical clarity of the membrane. Chemical crosslinking improved the light transmittance as did rehydrating dehydrated AM in normal saline, suggesting that these methods may improve patients’ visual outcomes in ophthalmic applications [83].

A simpler dehydration technique is air-drying, a less complex and more efficient method. The membranes are processed in a biosafety cabinet or laminar flow hood at room temperature and allowed to air dry for a defined period of time, usually followed by sterilization. This low-cost method results in a graft that can be stored at room temperature. A study by Singh et al. demonstrated air-dried hAM to be a promising wound dressing and impermeable to Bacillus, Escherichia coli, Pseudomonas, Citrobacter, Flavimonas and Staphylococcus [89]. Following AM preservation, to ensure the safety of the final product, many manufacturers choose to perform a sterilization step. Two predominant techniques are described in the following section.

### 3.3. Sterilization

In the United States, human allograft AM is regulated under section 361 of the Public Health Service (PHS) Act through the Food and Drug Administration (FDA) Center for Biologics Evaluation and Research (CBER) [90]. AM is considered a human cell, tissue, and cellular and tissue-based product (HCT/P), and as such, must follow 21 CFR 1271, which outlines the regulations for minimizing disease transmission [90]. Since there is an inherent risk of communicable disease transmission from transplanted birth tissue, several processes and regulatory controls have been established to mitigate this risk, including serological testing, microbial testing, and aseptic processing, as specified in the American Association of Tissue Banks (AATB) regulatory guidance [91]. Sterilization is a key step in reducing the risk of communicable disease transmission. The two most commonly used sterilization agents for AM are gamma (γ) irradiation and peracetic acid (PAA) [33,55,92].

Other methods of sterilization include the chemical sterilants ethylene oxide, glutaraldehyde, formaldehyde, ionizing irradiation, high-dose electron beam irradiation [93] or supercritical CO_2_ [94]. It is important for function, activity, and clinical applications to find a safe and appropriate dose of these agents. Singh et al. studied the chemical and barrier function characteristics of the amniotic membrane at different doses of γ-irradiation and demonstrated that there were no significant changes in water absorption capacity, microbial impermeability, FTIR chemical structure, and water vapor transmission rate across the range of 25–50 kGy doses. This study did not address cellular activity or specific structure integrity assessments of the membrane following sterilization doses [95].

Using irradiation in appropriate doses is known to be an effective procedure but has been found to interfere with the tissue structure. Various studies have demonstrated that γ-irradiation in higher doses can cause destruction and degradation of all three layers of AM [81,93,96]. Using lower 15–25 kGy doses has been shown to not affect the presence of growth factors or morphology in cryopreserved or lyophilized hAM [34,81,97]. Paolin et al. evaluated the effect of γ-irradiation on cytokine levels and the ultrastructure of the ECM of fresh-frozen and of freeze-dried hAM irradiated with 10, 20 or 30 kGy gamma radiation. Gamma radiation led to a significant loss of several cytokines and to increased damage to the epithelium and basement membrane with higher doses [81]. Djefal et al. performed a validation of a 25 kGy gamma radiation dose and confirmed that this dose met the ISO standard (ISO 11137) for a sterility assurance level (SAL) of 10^−6^ [97]. The other commonly used sterilization method, PAA, has been shown to preserve the typical structure of hAM compared to γ-irradiation [92]. PAA is a standard sterilizing agent that is highly effective against bacteria, viruses, and spores due to its high oxidizing potential and its non-toxic residuals [98,99]. Combinations of the different preservation methods with the optional sterilization methods impact the AM morphology, thickness, composition, and growth factor content.

Overall, several contradictory findings have been reported regarding the effects of preservation and sterilization methods on the levels of growth factors, storage retention, and the structural integrity of membranes. This ultimately comes down to the variation between donors and the lack of uniformity within processing and preservation steps performed after the initial collection of the donor tissue. Since every step of preparation, preservation and sterilization can influence the properties of a biological material (see Table 2), further studies are necessary to analyze the effects of procedures on the biological activities of hAM, hCM and hACM graft products. Although all of the above-mentioned factors influence AM features, it is evident that the different forms (decellularized, dried, and frozen) have unique properties suitable for many clinical applications. One way to address the variability among donors and the impact of processing and preservation methods on AM allografts includes chemical crosslinking.

### 3.4. Crosslinking

Despite the many favorable characteristics of AM allografts for tissue engineering it does have limitations. Allograft membranes are susceptible to native tissue collagenases that degrade the different types of collagens causing loss of structural integrity in the applied AM [34,100,101]. Crosslinking is an established method for stabilizing membranes and scaffolds and increasing their durability by decreasing the available sites for proteolysis. When fresh hAM is applied for soft tissue repair or regeneration it rapidly degrades, typically within one week, requiring reapplication, while cryopreserved hAM may remain intact up to several weeks [102,103]. Glutaraldehyde (GA), and 1-ethyl-3(3-dimethylaminopropyl) carbodiimide (EDC) have been commonly used in AM cross-linking [83,100,102,104]. Spoerl et al. demonstrated a significant increase in the biomechanical strength with increased resistance to degradation over a period of 7 days to 3 months in membranes cross-linked with 0.1% GA [102]. While GA has been used to crosslink a variety of biological materials, it must be completely removed or fully reacted prior to tissue implantation given its cytotoxic nature [101,103]. Sekar et al. used aluminum sulfate, Al_2_(SO_4_)_3_ to crosslink fresh hAM for use in corneal implantation and observed comparable tensile strength with fresh hAM, cytocompatibility with corneal limbal explants, sterility, and optical clarity, concluding that Al_2_(SO_4_)_3_ may be suitable for implantation studies [105]. Yi et al. conducted a comparison of GA and dialdehyde starch (DAS) to produce a contact lens-shaped, suture-less, crosslinked hAM with decreased degradation to alleviate patient discomfort associated with the existing O-ring-fixed hAM products. Among the several measures assessed, DAS-treated hAM demonstrated an intermediate increase in tensile strength and collagenase resistance between normal and GA-treated hAM and better retention of growth factors and cytocompatibility compared to GA. In a rabbit corneal model, DAS-treated hAM demonstrated a faster healing rate compared to untreated hAM. In a 7-day fit test in humans, five subjects who completed the study experienced secure attachment to the cornea and good tolerance was observed [106].

It is important to use a safe, effective, and optimal concentration of the crosslinking agent to ensure stability and minimal cytotoxicity effects. The cytotoxicity effects may depend on the concentration used in the reaction and, ultimately, how well the tissue is washed to remove residual crosslinker agents [100,101,103,107]. The varied methods for processing, preserving, sterilizing, and crosslinking lead to a wide array of hAM, hCM, hAAM, hACM and other multilayered amniotic membranes that are likely to fit diverse clinical niches and applications. In the following section, we describe four of the more common fields of clinical application utilizing AM for tissue repair and regeneration.

## 4. Clinical Applications

### 4.1. Introduction

The intended outcome of using regenerative materials for clinical applications is the acceleration of healing and the restoration of the supporting tissues that have been damaged or lost from surgery, trauma, disease, infection, or other complications. Over the past century, the use of amniotic membranes to support healing in the clinical setting has expanded from skin transplantation [7] and burns (skin and ocular) [9,10] to additional ophthalmic applications [50,108,109], periodontal conditions [110,111], acute and chronic wounds including diabetic foot ulcers, venous leg ulcers and pressure ulcers [75,112,113,114], to plastic and reconstructive surgery including first to third degree burns [19,115,116], post-Mohs wounds [117,118], and surgical adhesions [18,119,120,121]. A variety of characteristics such as the promotion of re-epithelialization, reduction in inflammation and fibrosis, immunomodulation, the inhibition of foreign body reaction, and antimicrobial activity make AM useful in clinical therapies [3,17,34]. The clinical application of amniotic membrane not only maintains the structural and anatomical configuration of regenerated tissues, but also contributes to the enhancement of healing through the reduction of postoperative scarring and subsequent loss of function [15]. When used as a dressing or wound covering, AM acts as a structural barrier facilitating hemostasis, reducing water loss, providing a barrier to microbial colonization and infection, and reducing pain [115,122,123]. The diversity of AM applications includes combination with various cell types or synthetic matrices such as collagens or alginates to create broad application-based regenerative products for healing [3].

The specific applications for amnion, chorion, amnion–chorion, and multilayered AM allografts are broad and have significant overlap. There is variability in application and outcomes between studies using hAM, hCM, hACM or multilayered hAM allografts, in part related to the types of applications, the different sizes of membrane used, which membrane surface will be in contact with the wound (stromal or epithelial side), folding of the membrane, multilayered usage, graft re-application, graft resorption and if any additional protective coverage was implemented when treating defects. From a simple barrier for topical skin-wound applications to corneal repair, to more advanced forms such as membrane extracts and flowable micronized membrane injections, clinical application of amniotic membranes has expanded and continues to evolve.

The purpose of this section is to highlight clinical applications of AM allografts in four prominent clinical fields: periodontal, ophthalmic, chronic wounds and plastics. The subspecialties provide descriptions of the different procedures used and why they are applied to AM within that clinical focus. The differences between membrane types within each subspecialty identifies which, if any, are considered optimal.

### 4.2. Periodontal and Oral Surgery

Periodontal and oral surgery are common interventions for a variety of periodontal complications and diseases. Depending on the specific disease presentation, a variety of periodontal surgical techniques can be utilized including flap surgery, bone grafting and implantation or soft tissue autografting. During these procedures oral tissue damage does occur, and AM products sufficiently address this. Since the mid-1990s, the use of hAM allografts in periodontal and oral surgery to accelerate tissue regeneration has expanded, showing promising results in various specialties of dentistry [28,62,110,124,125].

Amnion (hAM), hCM and hACM allografts are popular for periodontal indications, acting as scaffolds that promote cell adhesion and specific protein synthesis to support the growth of bone and gingival tissue for regenerative procedures. The thinness of hAM, while beneficial for application in tight spaces, can be difficult to handle for some procedures [111,124]. Conversely the thickness of the hCM and hACM allografts make them easier to handle and contain a higher amount of growth factors [57,84,111,124]. While there are no standardized practices described for the application of AM in the oral cavity, two general clinical procedures have been described [28,124]. The implant procedure involves placing hAM, hCM or hACM beneath the gingiva for guided tissue regeneration of gingival recession with coronally advanced flap repair, Figure 4, and guided bone regeneration [124,126,127,128,129]. It is also used for soft tissue repair following dental implant placement [130], intrabony defects [49], Schneiderian membrane perforation repair, [111,124], furcation defects [28,124] wound management following surgical implant or periodontal surgery, and bisphosphonate-related osteonecrosis of the jaw [28,111,124,126,127,128,129]. The overlay procedure describes the use of graft material as a covering or barrier in mucosal defects such as mandibular vestibuloplasty [110,131].

The oral cavity is populated by many microbes which can impact the success of surgical procedures. During oral surgical procedures, implanted membranes are colonized by resident microbes, leading to biofilm formation. Ashraf et al. identified antimicrobial activity in hACM against three bacterial species, comparable to a tetracycline positive control, and suggested that this function may improve outcomes in oral surgical applications [132]. Dehydrated hACM and hCM have been used successfully in a variety of periodontal and alveolar regenerative procedures [111,124,129]. The laminin content of the hAM and hCM tissue layers promotes cell adhesion and proliferation, a function that facilitates healing and improves the outcomes of periodontal and surgical procedures [28,62]. Depending on the specific procedure, the self-adherence of dehydrated AM enhances epithelialization, reduces surgical time, and may eliminate the need for sutures [62].

The combination of anti-inflammatory, antifibrotic, antimicrobial, low immunogenicity, growth factor and cytokine content, plus the elasticity of hAM or relative stiffness of hCM and hACM enhance their regenerative potential, making these allografts suitable for a wide range of oral treatments or applications. These attributes altogether make this material relatively easy to use and offer unique future treatment options.

#### Differences between Membrane Types in Periodontal Application

Several studies support the use of hAM, hCM and hACM as effective additions to the current materials and techniques utilized for periodontal and oral regenerative procedures. However, further studies are necessary to demonstrate their effectiveness and identify the strengths and weaknesses of each allograft type for the application. One case study compared the efficacy of hAM versus hCM for treatment of gingival recession using a coronally advanced flap procedure and found a significant increase in the width of keratinized gingiva and gingival thickness within 6 months in both treatment groups, providing promising results for root coverage. The small patient pool and 6-month follow up window limited the ability of the authors to draw definitive conclusions about which membrane would be optimal, leading to a recommendation for future longitudinal studies [128]. Gulameabasse et al. performed a systematic review of recent clinical applications of hCM and hACM, including the treatment modalities in oral and periodontal surgery for regenerative purposes and concluded that additional studies are required to distinguish the benefits of these membranes compared to PTFE and other resorbable, conventional membranes [111].

### 4.3. Ophthalmology and Ocular Surgery

To date, ophthalmology is one of the most established fields for the application of hAM. De Rotth was the first to describe the use of fresh placental membranes to repair conjunctival defects as a replacement for the patient’s mucous membranes [11]. In 1946, dehydrated hAM, termed “amnioplastin” was used to treat acute ocular burns, resulting in rapid recovery with few complications or sequelae [12]. Even with these initial successes, it is speculated that the lack of AM preservation methods may be the reason for the unreported use of AM in the treatment of ocular surface disorders until the early 1990s [2]. Since 1995, AM transplantation has been successfully applied to ocular surface reconstruction in patients with a variety of ocular surface diseases.

When the limbus, the location of the corneal limbal stem cells responsible for renewing the transparent corneal epithelium, is damaged, hAM provides a substrate for expanding limbal stem cells to regenerate and form new and healthy tissue [109,133]. When appropriately processed and preserved, AM can be used in three unique types of application for ocular surface disorders: graft, patch, or stuffing/sandwiching. When used to cover an area of the ocular surface, it is either used as a patch, also known as the overlay technique, or a graft, also known as the inlay technique (see Figure 5) [109]. A patch or overlay denotes the application of the AM as a biological bandage with the epithelial side against the defect, and it is sutured into place. The membrane will eventually be removed or will slough off. Epithelialization of the ocular surface typically occurs underneath the membrane/bandage, with either the stromal or epithelial side of the AM facing the defect. Graft or inlay application indicates that the AM is trimmed to the size of the defect with the stromal side against the defect, allowing the epithelial side (facing out) to act as a scaffold for re-epithelialization. The graft is incorporated into the host’s tissue [14,50,109,134]. When clinicians encounter deep corneal or scleral ulcerations or small perforations resistant to conventional medical therapies, they can be treated using the filling-in, stuffing, or sandwich techniques by placing small pieces of hAM and layering them into the stromal defects. These may be followed up with grafting or patching over the area with additional membrane or materials [14,50,135].

Amniotic membrane can be utilized for ocular surface reconstruction, including conjunctival repair and reconstruction, persistent corneal epithelial defects, corneal perforations, bullous keratopathy, limbal stem cell deficiency, deep corneal ulcers, neoplasia, and complications from Stevens–Johnson Syndrome [15,53,134,136,137]. The hAM can be clamped into a ring system. The fixation device application techniques have several advantages as they can be performed under topical anesthesia, so surgery time is shortened and there are no suture-related complications [108,138]. The orientation of the hAM is dependent on the application and clinical endpoint.

Ocular surface reconstruction applications have advanced considerably in the past two decades with the advent of amniotic membrane transplantation (AMT). In this tissue engineering technique, amniotic epithelial cells are harvested and expanded on denuded/decellularized amniotic membrane and transplanted to restore the structure and function of damaged ocular surfaces [50,109,133]. To restore the limbal cell population in limbal stem cell deficiencies caused by burns and other traumas using limbal cells from the healthy donor eye, simple limbal epithelial transplantation (SLET) or cultured (ex vivo) limbal epithelial transplantation (CLET) with hAM have been demonstrated to facilitate repopulation of the limbal stem cell niche [135,136,137,138,139,140,141,142].

When treating the ocular surface with hAM, suture fixation may be required to keep the membrane in place. If the ocular surface is heavily inflamed, the membrane disintegrates faster and may have to be reapplied several times or implanted with multiple layers, covered with a larger piece sutured in place [135,138]. Fixation with sutures adds additional surgical trauma for the patient, which can be avoided with suture-less applications [124,138]. Further improvements have been implemented to avoid hAM suturing and suture-removal entirely. The amnion can be fixed in place with a tissue adhesive such as fibrin glue or mounted on a plastic structure. For example, Prokera^®^ and AmnioClip are commercially available medical devices that provide a suture-less biological bandage of cryopreserved hAM clipped to a thermoplastic ring set [108,138,142,143].

The ocular surface is a unique combination of two phenotypically distinct epithelial populations. The corneal epithelium is derived from stem cells in the limbus and the bulbar conjunctival epithelium is derived from stem cells in the fornix and/or bulbar conjunctiva [53,144]. The immune-privileged properties of hAM and the corneal epithelium and limbal stem cells are similar enough to have been called “parallel universe” tissues [145]. When the limbus is damaged, bulbar conjunctival epithelial cells can migrate into the cornea, resulting in neovascularization, inflammation, and loss of visual acuity. Restoration of the limbus with the use of hAM and autologous or allogeneic limbal cells as described above is critical to the restoration of vision [139,140,141]. In severe ocular afflictions the damage may not be limited to the corneal and conjunctival epithelia alone; additional tissues may require reconstruction. Applying hAM with the epithelial side up and the stromal side in direct contact with the affected tissue provides a substrate for the corneal or conjunctival epithelial cells to proliferate on, facilitating repair and regeneration [15,50].

Azuara-Blanco et al. evaluated amniotic membrane transplantation (AMT) for treating persistent corneal epithelial defects (PED) and corneal ulcers using cryopreserved hAM and observed that AMT was effective at re-epithelializing PED unless there was severe epithelial thinning, which subsequently required limbal or tectonic transplantation to restore the epithelium [146]. Koizumi et al. observed that cultured, explanted rabbit corneal, central and limbal epithelial cells expanded more rapidly on denuded versus intact amnion and have uniform leading edges of cell expansion. They also found that corneal cells from the limbus colonized the amnion surface more readily than central corneal cells [72]. Rama et al. identified that limbal corneal epithelial cells initially cultured on fibrin to maintain their stem cell expression (holoclone) retained holoclone properties when cultured on intact amnion, leading to long term recovery in patients receiving the cultured amnion graft [139]. These studies depict the ability of hAM to encourage cellular proliferation and remodeling.

In a randomized parallel-controlled clinical trial of severe chemical injury, Eslani et al. compared standard medical care to AMT plus standard medical care, assessing the primary outcome of time to complete epithelialization. They observed no difference in time to epithelial healing or secondary outcomes: visual acuity, central corneal neovascularization, or symblepharon formation. Comparing their data with published clinical studies of AMT for treating acute ocular injury, they summarized that AMT may provide the initial benefit of pain relief but may not be needed in addition to medical care and this may be beneficial only in moderate cases. For severe injuries, AMT is insufficient at preventing secondary long-term complications [147].

The reintroduction of AM in ophthalmic surgery has been demonstrated to be a viable alternative for many clinically challenging situations; however, it is not a panacea for all ocular ailments. One important drawback reported by Dua et al. is the loss of the membrane, either by degradation or by cheese-wiring of the sutures causing an element of tension in the immediate post-operative period [15]. Another undesirable effect is the possible residual subepithelial membrane persisting and impairing the vision of the patient. This may happen if the membrane used is from a relatively thicker portion of the amnion, nearest the umbilical cord [15]. An accumulation of blood (hematoma formation) under the membrane can occur during the immediate postoperative period or during suture removal after transplant; blood is usually absorbed but may need to be drained if excessive. Despite the widespread use of hAM in ocular surgery, very few complications have been reported [14,15,109].

Amniotic membrane allograft has been shown to be a beneficial biological dressing in ophthalmology and provides three basic functions. First, it acts as a physical barrier and covering to protect the underlying surface and facilitate healing. Second, it provides a substrate to support re-epithelialization. Third, it is a source of anti-inflammatory and antifibrotic growth factors and cytokines which also foster epithelialization [38,108,148]. The use of hAM for ocular surface defects resulted in reduced neuropathic corneal pain [149] and reduced pain associated with the friction of the eyelids over the healing ocular surface [15,150]. The allograft is used in ocular repair to treat epithelial defects or ulcers, or as a barrier-like bandage to cover the ocular surface to promote healing or replace damaged avascular eye tissue. Its thin, lightweight, elastic and nearly transparent qualities make it a suitable material for use on eye surfaces.

#### Differences between Membrane Types for Ophthalmic Applications

The inherent properties of the human amniotic membrane for ophthalmologic indications are evolving and its use is being investigated to assess how or if it promotes faster healing compared to standard treatment. The literature regarding AM in the ophthalmology field describes the application of hAM without the chorion layer. This is due to the inherent relative transparency of hAM compared to the thicker and less transparent hCM and hACM. Currently, two types of hAM are commercially available for ocular use: cryopreserved and dehydrated. Both come in a variety of thickness and sizes, with or without rings, depending on clinical need.

### 4.4. Chronic Wounds

Chronic nonhealing wounds are defined as wounds that fail to progress through the systematic and timely healing process seen in an acute wound closure [151]. More specific definitions include wounds that remain open and unresolved for >one month [152,153] or ≥three months [154,155]. Common chronic wounds include diabetic foot ulcers (DFU), venous leg ulcers (VLU), arterial ulcers, pressure ulcers, ischemic ulcers, and surgical and traumatic wounds. These complex wounds create a substantial personal and economic burden to affected patients and a challenge for providers seeking alternative treatment to improve care and ultimately achieve complete and lasting wound closure [153,156]. Additionally, chronic, nonhealing wounds, often associated with advanced age or underlying health conditions such as obesity, vascular insufficiency, diabetes, poor nutrition, alcoholism, etc., are susceptible to infections which can lead to additional physician visits, hospitalization, osteomyelitis, or amputation [157,158,159].

Wound healing is commonly conceptualized as four overlapping phases: hemostasis, inflammation, proliferation, and remodeling. In a chronic wound, healing is arrested in the inflammatory or proliferation phases and is often exacerbated by confounding factors such as diabetes, smoking, malnutrition, cardiovascular disease, aging, obesity, and more [158]. Granulation tissue is diminished, and re-epithelialization is halted, resulting in a persistent open wound, subject to infection and additional complications. Healing a chronic wound requires a shift in the cytokines that perpetuate the arrested phase, including coordination of fibroblast and keratinocyte proliferation and migration. The growth factors and ECM of AM contribute to the rescue of a chronic wound, facilitating progression back into the normal healing phases.

DFUs are the most common complication of diabetes with a pooled global prevalence of 6.3% [160] and it was estimated 15% to 25% of diabetic patients will develop a DFU in their lifetime, increasing the risk of life-threatening comorbidities and complications such as infection, cellulitis, osteomyelitis, myocardial infarction, increased hospitalization, or amputation [159,161].

The treatment process for chronic wounds follows a standard of care (SOC), which employs procedures such as debridement (sharps, enzymatic, biological, etc.), controlling infection, dressings to maintain a moisture-controlled environment, compression, negative pressure wound therapy (VAC therapy), skin grafts and physical offloading of pressure sites, but despite the careful attention, some wounds remain unresolved [162,163].

Although AM is used in the standard treatment options of various ocular and ophthalmologic indications, it had not become a common standard of care for chronic wounds. Historically this is due to issues related to tissue preparation, storage, and sterilization; however, current screening and processing methods have eliminated most of these concerns. Today, the practice of utilizing AM allografts for chronic wound treatment is usually applied when conventional treatment options fail, and it is essential to reassess and modify the patient’s treatment therapy.

In a nonblinded case study, four patients with refractory nonhealing wounds were treated with dhACM following sharp debridement. Reapplication was required in two patients when wound healing plateaued. All patients achieved 100% wound closure within 3, 7, 8, or 17 weeks from the initial dhACM application [164]. This result warranted further investigation into the broader application of AM. In an alternative case series, Regulski treated four elderly patients with comorbidities, an age range of 69–85 years, with DFU, VLU or traumatic wounds that remained open for a minimum of four weeks, with SOC, viable hAM, and secondary dressing or multilayer compression therapy as appropriate. All wounds healed with one to eight applications over two to eight weeks [75].

Multiple studies have reported superior wound healing following the application of AM to chronic diabetic foot ulcers (DFUs), venous leg ulcers (VLUs) and refractory non-healing wounds of varying etiologies. The Zelen group performed the first randomized controlled prospective trial comparing dhACM with SOC for treating DFUs and observed that the average wound area reduction rate was 3-times higher after amnion administration than or SOC alone at four weeks [165]. A meta-analysis of data from seven prospective randomized controlled trials (RCTs) treating DFUs with dhAM, acellular hAM, dhACM and cryo-hACM was conducted. Treatment groups were amniotic membrane plus SOC compared to SOC alone, with outcome assessments at 4-, 6- and 12-weeks. Amniotic membrane plus SOC resulted in faster DFU healing than SOC alone [113].

A similar performance in patients with chronic VLUs was also observed in several studies. In a pilot study by Mermet et al., 15 patients with chronic VLU were treated with AM. Ulcer evaluations were performed at the time of transplantation, with local pain assessed using a 0–100 visual analog scale (VAS). The data demonstrated 100% graft-take, re-epithelialization from the wound margins, a significant reduction of ulcer-related pain within the first seven days of application, and a baseline ulcer surface area reduction in 80% of patients, with 20% experiencing complete healing during a 3-month follow-up period [114]. A multicenter RCT assessed the percentage reduction in VLU wound area at the fourth week of care as a surrogate for complete wound closure at 12 or 24 weeks for patients receiving one or two dhACM applications with multilayer compression therapy (MLCT) or MLCT alone. After four weeks of treatment, 62% of those receiving one or two applications of dHACM plus MLCT achieved >40% wound closure compared to 32% receiving MLCT alone. The four-week surrogate could not be confirmed given the lack of long-term follow up [166]. In a retrospective review, the investigators analyzed deidentified data from 101 patients with non-healing lower extremity wounds that did not show a 50% reduction in size over four weeks of SOC. Following sharp debridement, wounds were treated with dhACM in sheet form (52.5%), micronized form (14.8%) or both sheet and micronized forms (32.7%) for up to 12 applications within the 12-week period. The wounds in 92 (91.1%) of the patients healed and nine remained unhealed [159].

The application of amnion allografts plus SOC has resulted in significantly faster rates of wound closure compared to SOC alone for refractory non-healing wounds from a variety of etiologies, such as DFUs and VLUs [159,165,166,167,168]. With recent efforts focusing on optimizing effective strategies for treating chronic wounds, AM is transitioning into a priority treatment option.

In a retrospective analysis of two randomized clinical trials utilizing dhACM and dhUC (dehydrated human umbilical cord), from two placenta-derived allografts (PDAs), Tettelbach et al. revealed a correlation between effective SOC debridement and DFU wound closure rates. When adequate debridement was performed, 86% of patients receiving a PDA attained complete wound closure compared to 30% with inadequate debridement. For the control group 60% attained complete wound closure with adequate debridement versus 0% with inadequate debridement [169].

Chronic wounds have a compromised microvasculature that can result in hypoxia and apoptosis which prevent healing; however, AM has been shown to stimulate angiogenesis. As previously mentioned, recent data demonstrate a sidedness effect of angiogenic response in hAM, where the epithelial side is more antiangiogenic, and the stromal side is proangiogenic [39]. This appears to be different with hACM since the chorion layer has a higher concentration of proangiogenic factors. including but not limited to VEGF, fibroblast growth factor-basic (bFGF), and PDGF-BB [57,84,170]. In addition to a hypoxic environment due to impaired vascularization, the bias towards inflammation is in part impacted and perpetuated by the presence of proinflammatory cytokines, infiltration of inflammatory cells, proteolytic enzymes, microbial presence, and necrotic tissue. The diverse array of growth factors and cytokines present in AM appears to act as a homeostatic reset, facilitating the progression of a stalled wound into the remaining healing phases.

#### Difference between Membrane Types for Chronic Wound Applications

Dehydrated hACM is the placental membrane type used in many controlled clinical trials and case reports to date, followed by dhAM and cryopreserved hAM. There is a dearth of applications of hCM in chronic wounds: it is mostly used in periodontal applications as described earlier. Characterization of hCM continues which may result in clinical applications for chronic wounds in the future [44,69].

The efficacy of AM allografts for treating chronic wounds is actively being studied. Overall, current evidence suggests the use of AM in chronic wounds (especially DFUs) can achieve relatively faster wound closure rates than conventional treatment methods alone, with an overall reduction in healing time. While AM has shown positive effects on healing in chronic wound care, numerous studies have shown that the viable cells produce the factors that act to reduce inflammation, improve cellular proliferation, migration, and angiogenesis [171]. More clinical research is needed to decipher which allograft type and process provides the optimal potential for wound healing.

### 4.5. Plastic Surgery

Plastic surgery involves the restoration, reconstruction, or alteration of physical defects of form or function in the human body and can be divided into two main categories: reconstructive and cosmetic. Plastic and reconstructive surgery requires a conformable scaffold and dressing materials that adhere to and support a moisture controlled wound environment, prevents the formation of surgical adhesions, and facilitates vascularization of skin grafts, flaps, and other tissue repairs. Human amnion and chorion allografts provide the plastic surgeon with a versatile material that meets these requirements, with added benefits including the promotion of re-epithelialization with its unique extracellular matrix, presence of growth factors to accelerate healing, semi-permeability facilitating the exchange of gases and liquids, a barrier against debris and microorganisms, and painless application and removal [17,19]. Plastic surgery is a diverse field within medicine that offers many clinical usages for AM application. Plastics specialties described below include reconstructive surgery applications which include tissue grafting, tissue engineering, post-surgery reconstructions, and burn treatments with the use of AM.

AM is a biocompatible, flexible, conformable, and mechanically durable scaffold providing a covering or barrier. The first plastics applications of amnion were for the treatment of traumatic wounds, burns, chronic skin ulcers, and for skin grafting [7,8,9,10]. Skin grafts are a widely practiced technique with multiple plastic surgery applications including surface wounds, burns, mucosal lining repairs, split-thickness skin grafts, and flap repairs, and are often associated with a variety of complications, including creating a donor site wound, graft rejection, mismatches in mechanical properties, texture and color, hair growth, wound contraction, and scar formation. When treating large wound beds, either from trauma, surgery or other complications, AM has been used as an overlay following standard grafting to replace conventional dressings such as cadaveric allografts or xenografts [19]. Amnion applied to undersurface skin flaps has been shown to significantly increase angiogenesis, reduce infiltrating neutrophils, and improve overall skin flap survival [19,172]. Amnion used as a dressing material for flap donor site wounds contours well with the wound bed, is easy to apply, provides a protective barrier from wound contamination, reduces pain, and its translucency allows holistic flap monitoring, while acting as a microbial barrier and providing pain reduction by covering nerve endings [17].

AM has been used successfully in cranial surgery procedures, such as duraplasty following craniectomy [173,174] and craniotomy [16], and myelomeningocele surgery in combination with a sensate perforator flap [172]. It has been proposed that AM supports underlying neurological tissue through the production of neurotrophic factors, such as nerve growth factor [19,172]. Additional applications of AM for post-surgical interventions include the application of AM as a wrap for the prevention of adhesions in tendon surgery [121], for the ulnar nerve during cubital tunnel surgery [18], and for prostatic nerve bundle repair [120].

The unique extracellular matrix in hAM makes it a suitable scaffold for supporting various cell cultures, while ensuring safety, minimizing morbidity, and maximizing quality of life for the recipient. In tissue engineering, AM is used as a delivery system for growth factors, cytokines, and ECM as well as viable multipotent cells present in fresh and cryopreserved viable membranes. The AM basement membrane, whether intact or decellularized, has been used as a substrate for ex vivo or in vitro culture and for the expansion of cells to facilitate wound healing, e.g., limbal stem cells for ocular damage and keratinocytes and fibroblasts for wounds [19,30,138]. An in vitro living skin equivalent of normal human keratinocytes cultured on de-epithelialized hAM demonstrated a more robust basement membrane and healthier epidermis, suggesting this would be a suitable skin substitute for skin defects including treating burns, wounds, and ulcers [175].

There are situations where the natural biodegradation of AM matches the time to heal; however, many clinical procedures require reapplication or stabilization of the AM to facilitate complete healing. In these instances, crosslinking AM with GA or EDC/NHS has been shown to stabilize the membrane and enhance its resistance to proteolytic degradation [83,101,102,107].

The use of AM for patients who have undergone Mohs micrographic surgery (MMS) provides an alternative healing option to healing by secondary intention, reducing infection, and tissue scarring [118,176]. Mohs surgery is a procedure used in treating skin cancer and involves removing thin layers of skin and examining them closely for signs of cancer until cancer-free margins are achieved with minimal damage to healthy tissue. The process results in surgical defects of varying sizes, depths, and locations such as the lower extremities, scalp, conchae, face, and hands. After surgery, repair options, including secondary intention, are limited and the healing process may be protracted [118]. Patients undergoing Mohs with full-thickness defects exposing bone are more susceptible to healing impediments such as infection, pinpoint bleeding, dehiscence, and scar formation [117]. In a case series, five elderly patients, age range 72–98, received MMS resulting in surgical defects to the underlying bone. Dehydrated hACM allografts were applied directly on the bone weekly, resulting in the absence of pain and granulation tissue formation, and healing in seven to twenty-one weeks for three patients. The two remaining patients experienced wound healing, yet resolution was incomplete due to the presence of comorbidity [117].

For surgical procedures that require the removal of tissue, whether for cosmetic purposes or medical (such as Mohs), tissue is lost and a surgical defect can remain. A preliminary study assessed the efficacy of amniotic tissue-derived allografts on wound closure time and cosmetic improvement in Mohs patients. However, the study encountered a variety of limitations; both in the location of the wound and the initial defect size, which affected the wound closure rate. Further investigation is warranted into the beneficial use of amniotic allografts in postoperative Mohs care [118]. In a recent retrospective case study, a plastics and reconstructive surgeon utilized dhAAM to treat post-Mohs surgical defects in geriatric patients. Figure 6 demonstrates the rapid tissue granulation and improved wound closure time in a representative with a Mohs surgical defect on the scalp. The amnion graft applications encouraged rapid tissue granulation and improved wound closure time in a variety of Mohs surgical defects (See Figure 6). However, the study had limitations in sample size, wound variances, and patient demographics so statistically significant wound closure time was not captured [177]. In a larger-scale retrospective case-controlled study, 286 patients were assigned to control (autologous tissue flaps and full thickness skin grafts) or dhACM allografts and underwent MMS for basal or squamous cell carcinomas on the face, head, and neck. The patients who received the dhACM allograft had a five-fold lower incidence of infection and 97.9% had no postoperative complications compared to 71.3% in the autologous flaps and grafts group [178].

The application of hAM transplantation is effective both in the treatment of skin lesions and in patients suffering from toxic epidermal necrolysis (TEN) and Stevens–Johnson syndrome (SJS) [52]. TEN and SJS are classified as bullous diseases of the epidermis and manifest as severe cutaneous hypersensitivity reactions, with lesions affecting both the skin and mucous membranes and with poorly understood pathophysiology. Although TEN and SJS are rare, they are characterized by the high possibility of complications and a high mortality rate ranging from 3 to 10% for SJS and 25–50% for TEN [52,179]. More than half of patients being treated for SJS and TEN develop acute ocular involvement [136]. A 61-year-old patient with TEN covering 57% of her body, induced by treatment with an anti-epileptic, was treated with a two-fold treatment of intravenous immunoglobulin followed by amniotic membrane resulting in complete wound healing in 21 days [180].

The history of amnion usage in managing burn treatment is extensive, and compares its application with other treatment methods, with results indicating significantly reduced wound exudate, bacterial contamination, hypertrophic scarring, and healing time [181,182,183]. For many health care providers, the application of hAM has proven beneficial in the management and treatment of thermal wounds, from partial thickness burns to deeper and mixed burns [183]. The current treatment for burns involves early excision and grafting for deep partial-thickness and full-thickness burns. An RCT by Mohammadi et al. demonstrated a significant increase in the success rate of meshed split-thickness graft-take in burn wounds treated with fresh hAM dressings versus control [116]. In a controlled trial of dermal depth burns, Sawhney applied fresh hAM (up to 48 h from collection) or silver sulfadiazine cream control for three types of dermal burns: superficial, intermediate depth and deep. Both groups included bandage changes. In the hAM treatment groups, there was a significant decrease in time to heal for all three burn depth types with concomitant reductions in wound discharge and pain, and increased patient comfort during dressing changes [115]. A randomized pediatric partial-thickness facial burn treatment study compared frozen, glycerol-preserved amnion plus antimicrobial ointment versus antimicrobial ointment control on pediatric partial-thickness facial burns, and observed a significant reduction in dressing changes in the amnion group, but no significant changes in time to heal or the formation of hypertrophic scars at up to 12-months follow up, See Figure 7 for a demonstration of the healing of a facial burn using frozen, glycerol preserved amnion [181].

In a recent meta-analysis by Yang et al., eleven RCTs from 1985 to 2017 compared the efficacy of AM versus existing treatments, including SOC (three RCTs), silver sulfadiazine (three RCTs), polyurethane membrane (one RCT) and honey (one RCT) in acute burn wounds. AM was significantly more effective in reducing bacterial invasion, pain, scarring, and healing time compared to the existing treatments, except for honey, which was significantly more effective than AM. In skin-grafted burn wounds, AM was significantly more effective in reducing itching, scarring and time to graft-take compared with skin grafts affixed with staples [184].

A burn wound dressing needs to provide a basic physical and biological function, such as fluid handling, to promote the microenvironment conducive to wound healing. AM allografts used for temporary coverage of burn wounds exert both mechanical and physiological effects by protecting the wound, maintaining microbial control, covering exposed nerve endings, and hastening the healing process [115,122,123,181,182,183,185,186].

A common challenge for plastic surgeons in reconstruction surgery correcting facial and bodily defects from either birth, injury, disease, aging, or following surgery is maintaining cosmetic appearance. The human amniotic membrane provides an alternative treatment to improve the care for patients and achieve wound closure. Human amnion and chorion allografts provide the plastic surgeon with a versatile material that meets these requirements with some added benefits, such as promoting re-epithelialization with its unique protein matrix, allowing the elementary exchange of gases and liquids, acting as a barrier against debris and microorganisms, and painless application and removal. Its fluid handling capacity is a valuable function of a wound dressing, to cope with the exudates produced in vivo which are conducive to the microenvironment of a wound as it is healing [115,123,186]. As surgical techniques continue to advance, the effects of extensive scarring have diminished, but scarring is still possible.

#### Differences between Membrane Types for Plastic Applications

The unique properties of human amniotic membrane allografts are broadly used to heal and with the steadily evolving field of regenerative medicine, more applications are being investigated and implemented in plastics-based medicine. The two main AM types used for plastic and reconstructive surgery are hAM and hACM, fresh, dehydrated, and cryopreserved. Along with the intended uses and the subsequent clinical outcomes in plastics, continued controlled studies are needed to determine which AMs (hAM, hACM, fresh, dehydrated, cryopreserved) produce the most application-specific beneficial results for patients.

## 5. Conclusions

In this review, we describe the processing, preservation, and use of human AM from ethically sourced birth tissue for clinical application in the evolving and diverse field of regenerative medicine. The benefits of AM derive from the key components of the amniotic and chorionic membranes, including amniotic multipotent cells, thick basement membrane, compact, stromal, spongy, reticular and trophoblast layers. The cells and matrix components release a wide array of growth factors, cytokines, peptides, and soluble extracellular matrix components, which contribute to the observed tissue repair and regenerative effects.

Amniotic membranes have been prepared and preserved by a variety of methods with few standardized protocols; however, the fundamental components, ECM, and multipotent cells (viable or nonviable) have been demonstrated to positively impact a variety of clinical applications and outcomes. As continued understanding of these processing and preservation methods evolves, so will the regulatory requirements. The most investigated biological properties include immunomodulation, immunosuppression, anti-inflammatory, anti-scarring, analgesic proangiogenic, and antiangiogenic and antimicrobial.

There has been a considerable amount of research into the use of AM in a wide array of clinical presentations over the last century. Many fail to specify or describe the use of the different membranes, i.e., amnion, chorion, or multilayered amnion/chorion allografts. Even with this large evidence base, there is a paucity of well-designed, randomized controlled trials testing amnion performance against the gold standard alternatives such as SOC, autografts, xenografts, or synthetic tissue alternatives. Presently dhACM dominates the wound healing space; however, there are studies and applications using hAM, and to a more limited extent, hCM. There are insufficient data to generalize regarding the suitability of hAM, hCM and hACM products for most applications, with the exception of hAM use for corneal applications due to its transparency. Overall, the current literature provides insight into the immunomodulatory, anti-inflammatory, anti-scarring, and angiogenic potential of amniotic tissue products, further explaining the mechanisms behind the unique therapeutic capacities, but future studies must continue to further develop and understand the varying human placental tissue products. Additional controlled clinical trials and better standardized processing protocols will confirm safety and efficacy.

## Figures and Tables

**Figure 1 jfb-14-00531-f001:**
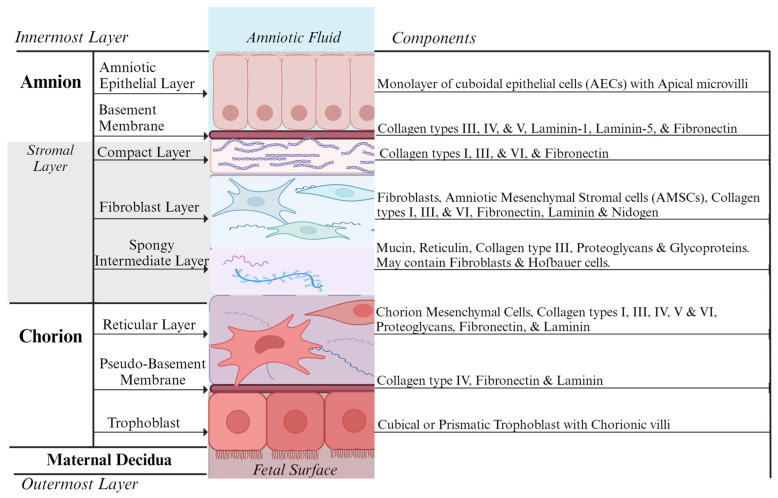
Structure and components of the amnion and chorion tissue layers. Created with www.biorender.com (accessed on 10 October 2023).

**Figure 2 jfb-14-00531-f002:**
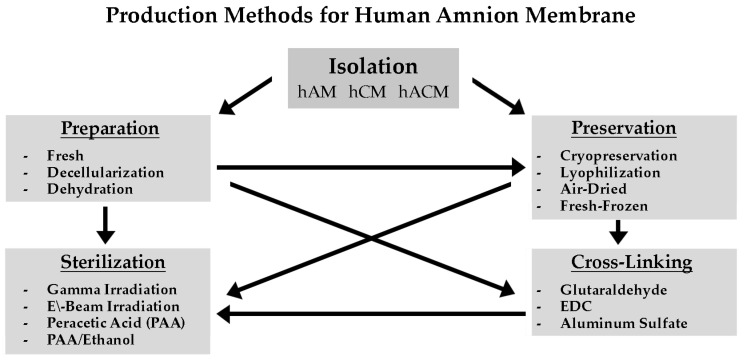
After isolation of amniotic tissue from human placenta, the membrane is dissected to separate the amnion membrane (hAM) from the chorion membrane (hCM), possibly folded or kept together as amnion–chorion membrane (hACM), for further processing. There exists a variety of ways for the membrane to be prepared, preserved, sterilized, or cross-linked for research or clinical use. The process is determined by the company and its specified membrane produced. Modified from Ref. [34] *Burns*, Vol 46(6), Gholipourmalekabadi M, et al., How preparation and preservation procedures affect the properties of amniotic membrane? How safe are the procedures? 1254–1271, Copyright (2020), with permission from Elsevier.

**Figure 3 jfb-14-00531-f003:**
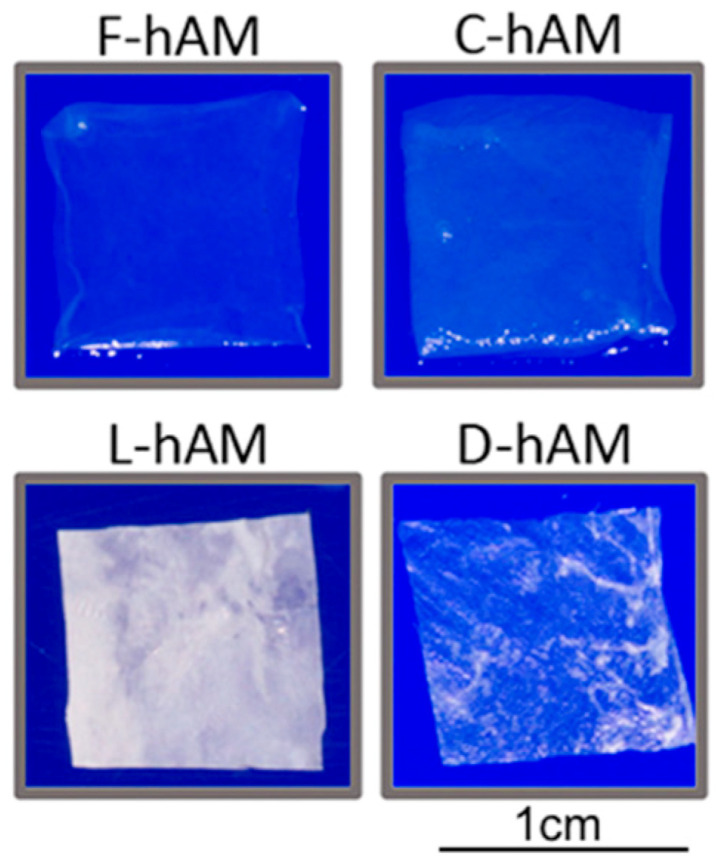
Comparison of the impact of preservation methods on amniotic membrane properties for tissue engineering applications. Human amniotic membrane formats: F-hAM, fresh, C-hAM, cryopreserved, L-hAM, lyophilized, and D-hAM, decellularized and lyophilized. Note the varying visual appearance based on preservation method. Reprinted from Ref. [65] *Materials Science and Engineering: C*, Vol 104, Fénelon M et al., Comparison of the impact of preservation methods on amniotic membrane properties for tissue engineering applications, 109903, Copyright (2019), with permission from Elsevier.

**Figure 4 jfb-14-00531-f004:**
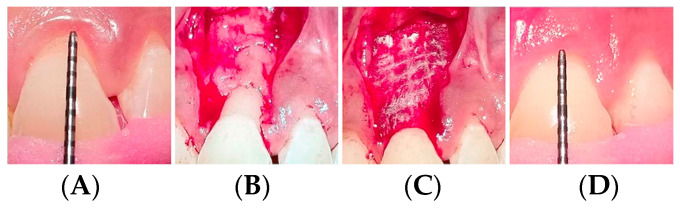
Example of gingival recession treatment using the coronally advanced flap technique to expose the root. (**A**) Baseline photograph; (**B**) flap reflection and recipient bed preparation; (**C**) amnion membrane placement; (**D**) six months postoperative. Reprinted from Ref. [126] the *Journal of Clinical and Diagnostic Research*, 11(8), Jain A, et. al., Comparative Evaluation of Platelet Rich Fibrin and Dehydrated Amniotic Membrane for the Treatment of Gingival Recession- A Clinical Study, ZC24-ZC28, 2017. Copyright Creative Commons License, https://creativecommons.org/licenses/by-nc-nd/4.0/ (accessed on 13 October 2023).

**Figure 5 jfb-14-00531-f005:**
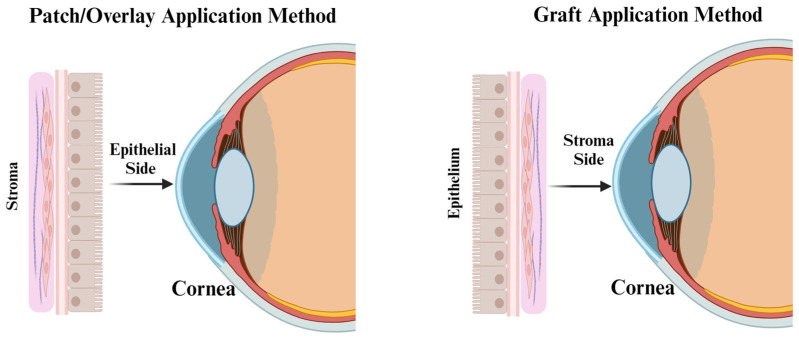
hAM patch and graft ocular application: when used as a patch (overlay), hAM is sized to cover the defect and sutured into place with the stromal or epithelial in contact with the defect. When used as a graft (inlay), hAM is trimmed to the size of the defect and is placed epithelial side up (stromal side in contact with the cornea) to facilitate epithelialization. Created with www.biorender.com (accessed on 4 October 2023).

**Figure 6 jfb-14-00531-f006:**

Example of a left forehead Mohs micrographic surgery defect treated with dhAAM. The image on the far (**left**) shows the patient’s initial wound exam before treatment. The (**middle**) image displays 3-week follow-up post treatment, with the last image (far (**right**)) depicting complete wound closure by week five. Reprinted from Ref. [177] *Journal of Medical Case Reports and Case Series* 4(17): Ingraldi AL, Lee D, Tabor AJ (2023) Post-Mohs Surgical Defect Repair with Dehydrated Human Amnion-Amnion Membrane: A Retrospective Clinical Case Study.

**Figure 7 jfb-14-00531-f007:**
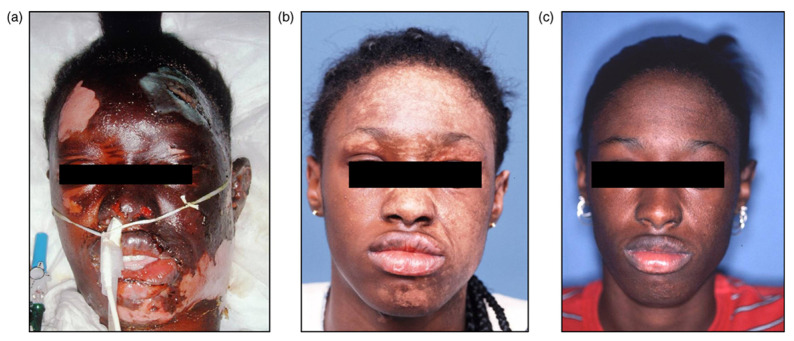
Long-term photographic results—amnion. (**a**) Fifteen-year-old female patient with partial-thickness facial burns after a house fire. (**b**) Follow-up pictures at intermediate timepoint. (**c**) 1 year follow-up shows complete healing and re-pigmentation. Reprinted from Ref. [171] *Burns*, Vol 34(3), Branski LK et al., Amnion in the treatment of pediatric partial-thickness facial burns, 393–399, Copyright (2008), with permission from Elsevier.

**Table 1 jfb-14-00531-t001:** Properties and functions of the amniotic membrane.

Properties	Contributing Factors	References
Anti-inflammatory	AM suppress the pro-inflammatory cytokines such as TNFα, IL-1, IL-6 and IL-8 and produce anti-inflammatory factors: IL-10, IL-4, TGF-β, HGF, PGE-2, HLA-G, and IDO.	1, 4–6,38, and 76,
Anti-microbial	AM serves as physical barrier against the external environment with close adherence to wound surface and producing anti-microbial peptides such as beta defensins, elafin.	1, 30, 41, and 44–46
Anti-scarring	AM reduces MMP and other proteases via the secretion of TIMPs, and downregulation of TGF-β.	1, 76, and 88
Non-immunogenic/low antigenicity	Low expression of histocompatibility (HLA Class II) antigens A, B, C, DR or β2. Presence of HLA-G and Fas ligand.	43, 50, 71, and 121
Analgesic properties	Pain relief is proposed due to efficient covering of the nerve endings. Anti-inflammatory growth factors such as IL-10, IL-1RA proposed to contribute to pain relief.	2, 6, 183 and 185
Angiogenic	Pro-angiogenic factors observed: VEGF-A, angiopoietin-1, HGF, and FGF-2, PEDF, MMPs.Anti-angiogenic factors: TSP-1, endostatin, TIMPs 1, 2, 3, and 4.	4, 5, 39, 40, 57, 76, and 169
Promote cellular differentiation and adhesion	Contains the structural proteins: Collagen types I through VI and VII, laminin, fibronectin, and vitronectin.	22, 28, 30, 35, and 62
Supporting epithelialization	Basement membrane is a substrate for cell migration, proliferation differentiation, and epithelialization with growth factors such as: KGF, b-FGF, and TGF-β.	1, 65, 78, and 153

**Table 2 jfb-14-00531-t002:** Comparison of production methods for processing hAM, hCM and hACM allografts.

Production Method	Storage	Membrane Morphology	Growth Factor and Protein Content *	Clinical Application
Fresh-Frozen	Immediate transplantation; Short-term cold storage (+4 °C)	Intact—depending on processing technique and handling	Preserved and ordinary GF and protein profile—may vary with donor	Thaw and apply or washed with saline then readily applied
Decellularization	Depends on preservation method applied	Damage membrane integrity, decrease in membrane thickness	Promotes greater cell proliferation, differentiation, and migration. GF content change observed	Depends on preservation method applied
Cryopreservation	Store and transport at −80 °C	Intact structural integrity, membrane thickness varies	Loss of certain growth factors and proteins (changes depending on if decellularized or not)	Thaw and apply, may apply with sterile saline
Lyophilization	Long-term room temperature storage	Maintains morphological structure	Decreased protein content and growth factor concentrations	Apply dry or wet (w/sterile saline)
Temperature-Controlled Dehydration	Long-term room temperature storage	ECM may be compromised	Retains active growth factors, lower levels of proteins	Apply dry or wet (w/sterile saline)
Room Temperature Dehydration	Long-term room temperature storage	May thin and increase membrane fragility	Maintains growth factor profile	Apply dry or wet (w/sterile saline)

* The inclusion or exclusion of layers of the amnion tissue within the created graft may affect the growth factor and cytokine content as well as the graft structure. Donor variability may also apply.

## Data Availability

No new data were created or analyzed in this study. Data sharing is not applicable to this article.

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
