# Peer review of "The Preparation and Clinical Efficacy of Amnion-Derived Membranes: A Review"

_jfb, 2023, doi:10.3390/jfb14100531_

Round 1

Reviewer 1 Report

In general, subsections should be highlighted with numbers or black colours

Section 2: It would be easier to follow if subsections are included

Figure 1: Please, indicate the source of the figure

Tables 1 and 2: no references are included. Is it own elaboration? Where did the authors get the information?

Section 4: More images should be included for each application

Figure 4: Please improve the image. It is very difficult to understand the differences between patch and graft.

English grammar is good in general with the exception: "The historic operations stimulated research and clinical investigations into exploring additional human therapeutic utilizations. As research supported the use of AM for anti-inflammatory, angiogenic, anti-angiogenic and antimicrobial properties, and a source of diverse growth factors, more clinically relevant applications have been demonstrated". Please, rewrite it.

Reviewer 2 Report

The manuscript entitled “The Clinical Efficacy of Amnion-Amnion and Amnion-Chorion Allograft Membranes: A Review” has been well prepared, however, there are several areas that need to be revised and strengthened the work, may authors consider getting the work better. In general, I did not observe any fundamental and serious issues in this work, and I recommend publishing this work after considering the major comments.

1.     The naming of third author is not journal’s standard.

2.     Based on MS content, it’s better to alter title to fit the content and clearly represent the paper topic. Avoid acronyms in the title.

3.     Abstract must be enriched by the important results of the study.

4.     Keywords should reflect the key points of the work. Be corrected.

5.     Novelty of the work should be highlighted in the introduction.

6.     The review of the literature needs more updating with works to have a clear and concise state-of-the-art analysis. This should more clearly show the knowledge gaps identified and link them to the paper's goals. The major defect of this study is the debate or argument is not clearly stated. The relevant reference may be of interest to the author according to below:

https://doi.org/10.1016/j.apmt.2022.101532

7.     All the tables and figures have no references.

8.     The description of Figure 1&2 provide valuable information about the structure and composition of the materials studied. However, it lacks critical analysis and interpretation of the observed features. It would be beneficial to include a more in-depth discussion of the significance of these findings, how they relate to the study's objectives, and potential implications for the research. This would help readers better understand the relevance of the presented data within the context of the study.

9.     Please avoid having one heading after another with no discussion in between as in the case of Sections 2 and 2.1. Kindly inspect the entire document for similar instances and revise accordingly. Please add in the beginning of your scientific hypothesis. In the course of describing the performed actions, please provide reader guidance, sufficient for understanding why those actions have been performed. The percentage purity and company of all reagents/chemicals utilized must be reported. Though some of the model/brands of the equipment used was stated, their country of manufacture should be reported as well.

10.  All the findings of the current work need to be compared and discussed with the results of other researchers finding instead of having a general comparison with other researchers' works. The authors should perform a comparison between the forecasting results. In your discussion section, please link your empirical results with a broader and deeper literature review. In order to make benefit from the superiority of the present work and help the audience and readers to understand the value of the research, the comparison of the results of the present work with previous works should be reposted as an individual Table.

11.  The conclusion is not acceptable in its present form. Therefore, it must be improved by more data. Please make sure your conclusions section underscores the scientific value-added of your paper, and/or the applicability of your findings/results. Highlights the novelty of your study. A brief restatement of your hypotheses. Your vision for future work. In the conclusions, in addition to summarizing the actions taken and results, please strengthen the explanation of their significance. It is recommended to use quantitative reasoning compared with appropriate benchmarks, especially those stemming from previous work.

12.   The reference list should be updated with recent publications, as some of the cited references appear to be outdated. Please check them carefully and correct the inconsistency.

13.  There are several grammatical errors in the manuscript. So, the language should be polished throughout the manuscript before publishing.

Reviewer 3 Report

The aim of this manuscript is to provide a background on the anatomical and physiological properties of amniotic membrane, the types of medically available products including the varying AM processing methods, and a review of varying current clinical uses.

This manuscript shows rich content, providing a deep insight for some works: the study is within the journal’s scope, and I found it to be well-written, providing sufficient information. Even if the manuscript provides an organic overview, with a densely organized structure and based on well-synthetized evidence, there are some suggestions necessary to make the article complete and fully readable. For these reasons, the manuscript requires major changes.

Please find below an enumerated list of comments on my review of the manuscript:

The authors should provide a list of the abbreviations mentioned in this manuscript.

ABSTRACT:

LINE 24: The authors should reformulate this sentence as following: “ The aim of this manuscript is to compare the various properties of human allograft membranes and to discuss their use in tissue engineering and patient clinical response”.

INTRODUCTION:

LINE 33: The distinctive structure of AM, linked to its biological and physical characteristics, make it a highly biocompatible material in a variety of regenerative medicine applications (see, for reference: https://doi.org/10.1016/j.mtbio.2023.100790). In this introductive section, the authors should mention how the biological and physical properties of AM contribute to its application in multiple regenerative contexts.

DISCUSSION:

LINE 521: Regenerative medicine have enhanced the development of several classes of biomaterials, applied in different regenerative contexts, specifically in periodontally regenerative procedures. The main hallmark of scaffold materials is to induce cell adhesion, enhance specific protein synthesis, and support the growth of bone tissue, as suggested by recent scientific evidence (see, for reference: Bianchi, S.; Bernardi, S.; Simeone, D.; Torge, D.; Macchiarelli, G.; Marchetti, E. Proliferation and Morphological Assessment of Human Periodontal Ligament Fibroblast towards Bovine Pericardium Membranes: An In Vitro Study. Materials 202215, 8284. https://doi.org/10.3390/ma15238284). This is the major concern of this manuscript: the authors should highlight the most important contribute of biomaterials, to periodontally regenerative procedures, as confirmed by recent scientific evidence on this topic.

The main topic is interesting, and certainly of great clinical impact. As regards the originality and strengths of this manuscript, this is a significant contribute to the ongoing research on this topic, as it extends the research field on the anatomical and physiological properties of amniotic membrane, the types of medically available products including the varying AM processing methods, and a review of varying current clinical uses. Overall, the contents are rich, and the authors also give their deep insight for some works.

Furthermore, there is a specific and detailed explanation for the scientific evidence mentioned in this study: this is particularly significant, since the manuscript relies on a multitude of analysis, to derive its conclusions. Finally, the results are adequately discussed.

The conclusion of this manuscript is perfectly in line with the main purpose of the paper: the authors have designed and conducted the study properly. As regards the conclusions, they are well written and present an adequate balance between the description of previous findings and the results presented by the authors.

Finally, this manuscript also shows a basic structure, properly divided and looks like very informative on this topic. Furthermore, figures and tables are complete, organized in an organic manner and easy to read, even if the authors should improve the quality of Figure 1.

In conclusion, this manuscript is densely presented and well organized, based on well-synthetized evidence. The authors were lucid in their style of writing, making it easy to read and understand the message, portrayed in the manuscript. Besides, the methodology design was appropriately implemented within the study. However, many of the topics are very concisely covered. This manuscript provided a comprehensive analysis of current knowledge in this field. Moreover, this research has futuristic importance and could be potential for future research. However, major concerns of this manuscript are with the introductive and discussive sections: for these reasons, I have major comments for these sections, for improvement before acceptance for publication. The article is accurate and provides relevant information on the topic and I have some major points to make, that may help to improve the quality of the current manuscript and maximize its scientific impact. I would accept this manuscript if the comments are addressed properly.

Minor editing of English Language is required.

Reviewer 4 Report

no suggestions

Round 2

Reviewer 1 Report

The authors addressed my comments . It would be good if the authors include one image in the section 4.1. Periodontal and Oral Surgery. This referee understand that the authors rea waiting to obtain permissions but, in my opinion, it is worth to wait until permissions are granted

Reviewer 2 Report

There are some new studies of materials about diabetic, chronic wound healing with appropriate physicochemical and swelling properties to accelerate wound healing. Also, artificial small-diameter blood vessels (SDBVs) are extremely limited in their thrombosis and still present significant clinical challenges while healing wounds.

Reviewer 3 Report

The authors significantly improved the manuscript.
